# Downscaling Land Surface Temperature Based on Non-Linear Geographically Weighted Regressive Model over Urban Areas

**Shumin Wang [1] , Youming Luo [1], Xia Li [1], Kaixiang Yang [1], Qiang Liu [1,2,*] , Xiaobo Luo [3] and Xiuhong Li [1,2]**

1   College of Global Change and Earth System Science, Beijing Normal University, Beijing 100875, China; shumin_wang@mail.bnu.edu.cn (S.W.); luoyouming@mail.bnu.edu.cn (Y.L.); janelixia613@163.com (X.L.); 201921490044@mail.bnu.edu.cn (K.Y.); lixh@bnu.edu.cn (X.L.)
2   State Key Laboratory of Remote Sensing Science, Beijing Normal University, Beijing 100875, China
3   Institute of Computer Science and Technology, Chongqing University of Posts and Telecommunications, Chongqing 400065, China; Luoxb@cqupt.edu.cn
*   Correspondence: toliuqiang@bnu.edu.cn

**Abstract:** Land surface temperature (LST) is a vital physical parameter in geoscience research and plays a prominent role in surface and atmosphere interaction. Due to technical restrictions, the spatiotemporal resolution of satellite remote sensing LST data is relatively low, which limits the potential applications of these data. An LST downscaling algorithm can effectively alleviate this problem and endow the LST data with more spatial details. Considering the spatial nonstationarity, downscaling algorithms have been gradually developed from least square models to geographical models. The current geographical LST downscaling models only consider the linear relationship between LST and auxiliary parameters, whereas non-linear relationships are neglected. Our study addressed this issue by proposing an LST downscaling algorithm based on a non-linear geographically weighted regressive (NL-GWR) model and selected the optimal combination of parameters to downscale the spatial resolution of a moderate resolution imaging spectroradiometer (MODIS) LST from 1000 m to 100 m. We selected Jinan city in north China and Wuhan city in south China from different seasons as study areas and used Landsat 8 images as reference data to verify the downscaling LST. The results indicated that the NL-GWR model performed well in all the study areas with lower root mean square error (RMSE) and mean absolute error (MAE), rather than the linear model.

**Keywords:** land surface temperature (LST); non-linear geographically weighted regressive (NL-GWR); spatial downscaling; indices selection



## 1. Introduction

Land surface temperature (LST) is one of the most important parameters for climate research and has been widely used in many research fields [1,2], including urban heat island monitoring [3], hydrological cycle [4], and climate change assessment [5], etc. In recent years, the spatial heterogeneity of land cover in urban areas has been drawing more and more research interest, which requires satellite remote sensing LST data with a higher spatiotemporal resolution [6]; however, there are many difficulties in acquiring high spatiotemporal data. According to the spatial and temporal resolution, the current remote sensing data can be divided into two categories. One is the data with finer temporal resolution and coarser spatial resolution. For instance, the temporal resolution of the FY-4/AGRI satellite is 15 min/full replay, but its spatial resolution of the thermal infrared band is 4 km; the temporal resolution of the Himawari-8/AHI is the same as FY-4/AGRI and its spatial resolution of the thermal infrared band is 2 km [7]; both the advanced very high-resolution radiometer (AVHRR) and the moderate resolution imaging spectroradiometer (MODIS) have a temporal resolution of twice a day, and the spatial resolution of thermal infrared bands are 1.1 and 1 km, respectively [8,9]. The other data category has finer spatial resolution and coarser temporal resolution. The Landsat satellite

thematic mapper (TM), enhanced thematic mapper plus (ETM+), and thermal infrared sensor (TIRS) have a spatial resolution in thermal infrared bands of 120, 60 and 100 m, respectively, whereas their temporal resolutions are about 16 days; the imaging frequency of advanced spaceborne thermal emission and reflection radiometer (ASTER) is 16 days and the spatial resolution of its thermal infrared band is 90 m [10–12]. In general, a spatial resolution of 100 m and the temporal resolution of once or twice per day is desirable for research on urban heat source distribution; however, the spatiotemporal resolution of satellite remote sensing data cannot achieve this requirement [13]. There are two solutions capable of improving the spatial resolution of thermal infrared images. One solution is to improve the hardware, e.g., improving detector fabrication to increase the spatiotemporal resolution of the satellite remote sensors; however, this would have high production costs and take a significant amount of time. The second solution would be to improve image processing, which uses the visible-near infrared band of higher resolution data and selects an appropriate algorithm to improve the spatial resolution of the thermal infrared band [14]. LST disaggregation algorithms are widely applied to improve spatial resolution by unmixing the thermal infrared pixel. LST disaggregation is divided into two categories: temperature unmixing and thermal sharpening. The temperature unmixing means unmixing the LST of components while maintaining a constant spatial resolution of the pixels. Thermal sharpening refers to improving the spatial resolution by enhancing the spatial details of LST. Among these solutions, LST downscaling belongs to the category of thermal sharpening [15]. During the development of these algorithms, LST downscaling has branched out the following aspects: image fusion model, statistical linear regression model, modulation model, and hybrid model.

The downscaling algorithms based on the image fusion model obtain higher spatial resolution data with continuous time series by fusing the remote sensing data with different temporal and spatial resolutions. Fasbender et al. [16] proposed a Bayesian fusion method, which was proved to have a high accuracy and practicality when fusing visible and near-infrared bands of ASTER together. The spatial and temporal adaptive reflectance fusion model (STARFM) was proposed by Gao et al. [17], and achieved good results in reflectivity image fusion; however, when applied to LST data, the STARFM can only increase the spatial resolution of the effective LST data and is not continuous in the time series. Zhu et al. [18] put forward the flexible spatiotemporal data fusion (FSDAF) model, which required fewer auxiliary parameters to achieve a high prediction accuracy when there is little change in land covers. In the past few years, with the progress and development of artificial intelligence, machine learning methods were applied to image fusion by scholars. Bindhu et al. [19] proposed the backpropagation (BP) neural network to fit the non-linear downscaling non-linear DisTrad (NL-DisTrad) algorithm, which achieved excellent downscaling results when applied to estimating evapotranspiration. Dong et al. applied the neural network to image super-resolution reconstruction, then improved the super-resolution convolution neural network (SRCNN) model and proposed the accelerating the fast super-resolution convolutional neural network (FSRCNN) model, which can input low-resolution images into the network directly [20,21]. Ledig et al. [22] proposed the generative adversarial network (GAN), which has been widely applied to research. Recently, some researchers improved this model, including the dilated convolution generative adversarial network (DCGAN), Wasserstein generative adversarial network (WGAN), and boundary equilibrium generative adversarial network (BEGAN) [23–25]. Shao et al. [26] verified the GAN in thermal infrared image reconstruction. The results showed that this method could improve the image's subjective visual effect while maintaining objective quality evaluation.

The algorithms based on the modulation model are used to distribute LST with a coarse resolution to each sub-pixel proportionally. Guo et al. [27] summarized past studies and proposed the pixel block intensity modulation (PBIM) algorithm, which selects the panchromatic band as the scale factor. The PBIM algorithm can retain the original thermal spectrum information and integrate detailed information into the thermal infrared band.

Stathopoulou et al. [28], based on the PBIM method, used emissivity and two LST data types to simulate LST with high resolution to downscale the LST and achieved better results.

The algorithm based on a hybrid model utilizes many types of models. Zhukov et al. [29] proposed an multisensor multiresolution (MMT) method based on spectral decomposition to downscale the TM data and used the ASTER data to verify the downscaling results. Deng et al. [30] applied the spectral unmixing and thermal mixing (SUTM) algorithm to downscale the LST in urban areas.

Due to easier performance and better efficiency, the LST downscaling algorithms based on statistical models have been widely used in recent research. These algorithms can be divided into three types. The first one builds a statistical model between LST and NDVI to achieve the LST downscaling. The second one considers the complexity of the land covers and use variety auxiliary parameters for LST downscaling. The third one considers the spatial non-stationarity between LST and auxiliary parameters and applies the geographical model instead of the global model to downscale LST. These three types of models are under the assumption of scale invariability.

Kustas et al. [31] proposed disaggregation procedures for the radiometric (DisTrad) algorithm, which constructs a relationship between LST and NDVI to downscale the MODIS LST spatial resolution from 1000 to 250 m. Agam et al. [32] improved the DisTrad algorithm and proposed an algorithm for sharpening thermal imagery (TsHARP). This algorithm considered the non-linear relationship between LST and NDVI by adding the $NDVI^2$ into the regression relationship. Yang et al. [33] pointed out that a single factor cannot reflect the difference of LST in different land covers and considered building a regressive relationship between LST and multiple factors. The downscaling results showed that this algorithm was better than DisTrad and indicated that selected multiple parameters would better explain LST. Zhu et al. [34] proposed an improved hierarchical regression method and compared it with DisTrad and TsHARP algorithms, and results showed that this method had obvious advantages and obtained higher accuracy than the other algorithms. Eswar improved the DisTrad algorithm to select NDVI, fractional vegetation cover (FVC), normalized difference water index (NDWI), and soil-adjusted vegetation index (SAVI) as the auxiliary parameters, and the results showed that the FVC and NDWI had higher accuracy for humid areas, whereas the NDVI was more suitable for dry areas [35]. Stefania et al. [36] considered the NDVI, SAVI, normalized difference built-up index (NDBI), and urban index (UI) parameters and selected different parameter combinations for LST downscaling, obtaining the optimal parameter combination for LST downscaling. Qi et al. proposed a new method to combine multiple variable and machine learning algorithms to downscale LST in urban areas [37]. Wang et al. compared and analyzed the downscaled results based on multiple linear regression (MLR), TsHARP, and random forest (RF) methods and indicated that the RF model is applicable to downscaling research in heterogeneous regions [38]. The above downscaling algorithms for LST are examples of global models that build relationships with a global scope and assume stable relationships between LST and auxiliary parameters; however, the non-stationarity relationship between independent variables and dependent variables was not considered. Researchers have paid attention to the non-stationarity between LST and auxiliary parameters and proposed geographical models to downscale LST in recent years. Duan et al. [39] proposed LST downscaling algorithm based on the geographically weighted regressive (GWR) model, which was the first time a geographical model was proposed to downscale LST and achieved better downscaling results than the global TsHARP algorithm. After that, scholars have made improvements to the GWR model. Pereira et al. [40] proposed the geographically weighted regressive kriging (GWRK) model, which downscales the LST of ASTER; Peng et al. [41] considered both the spatial and temporal non-stationarity and proposed the geographically temporally weighted regressive (GTWR) model, which was compared with the GWR and TsHARP models and obtained better downscaling results; Wang et al. [42] considered spatial non-stationarity and autocorrelation simultaneously and proposed the geographically weighted autoregres-

sive (GWAR) model to downscale the spatial resolution of MODIS LST from 1000 to 100 m and achieved good downscaling results.

Based on the above, the statistical downscaling models evolved from a linear model to a geographical non-linear model, added different auxiliary parameters, and considered the non-stationarity relationship between LST and auxiliary parameters. However, the research into GWR and its improved models only considered the geographical linear relationship between LST and auxiliary parameters; the geographical non-linear relationship was overlooked. In this study, we addressed the complexity of the land covers in urban areas, and proposed a non-linear geographically weighted regressive (NL-GWR) model to downscale the MODIS LST.

## 2. Study Area and Data Preparation

### 2.1. Study Area

In this study, we selected two Chinese provincial capital cities, Jinan and Wuhan, as the study areas. Figure 1 shows the false-color images generated from Landsat 8 data of the study areas.

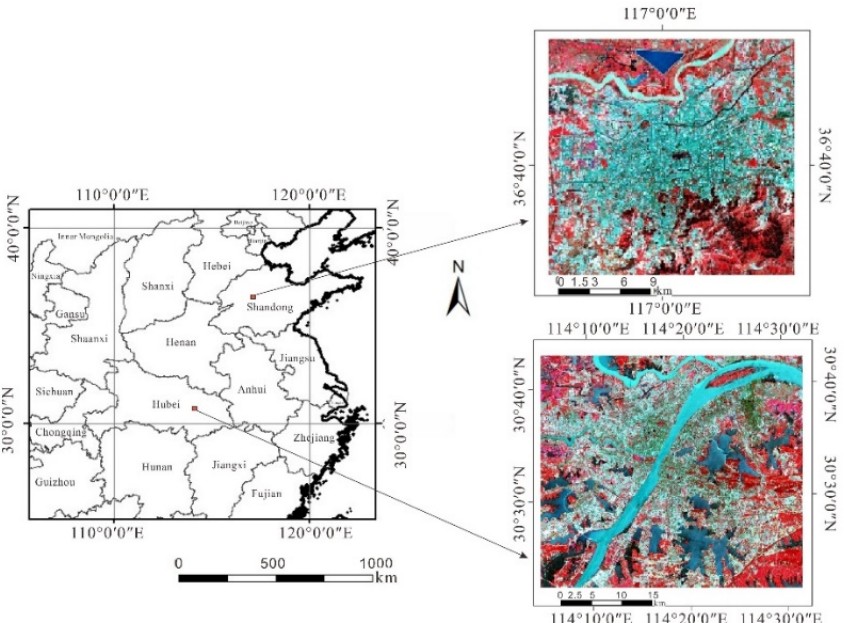

**Figure 1.** Landsat 8 false-color images (R: band 5, G: band 4, B: band 3) in Jinan and Wuhan.

Jinan is a typical city in the North China Plain. The selected study area covers a total area of 7998 km$^2$. Jinan has a warm temperate continental monsoon climate with a cold winter and hot summer. The annual average temperature is approximately 13.8 °C, where the lowest and highest temperatures are −19.7 and 42.5 °C, respectively. The average annual precipitation is approximately 685 mm. The study area selected in this paper is the Jinan urban area, and the main types of land cover include building, vegetation, water, and bare soil.

Wuhan covers a total area of 8569 km$^2$. Wuhan represents cities in the mid-south of China where rivers and lakes are widely distributed. It has a subtropical monsoon humid climate, which has abundant rainfall, sufficient heat, and four distinct seasons. The weather is hotter and wetter than Jinan, with an annual average temperature of about 15.8 to 17.5 °C, and the annual precipitation is 1150–1450 mm. The land covers are similar to Jinan.

### 2.2. Datasets and Preprocessing

The remote sensing data used in this study include Landsat 8 raw reflectance data and MODIS LST data. Three pairs of images were collected for each study area to test

the algorithm's performance in different seasons. Table 1 shows the main characteristics of these images in Jinan and Wuhan. Table 1 indicates that the Landsat and MODIS images were acquired within half an hour time. Due to the short time difference, the solar geometries, orbital parameters, and the viewing (near-nadir) of the MODIS Terra platform are highly consistent those of the corresponding Landsat 8.

**Table 1.** The Landsat 8 and moderate resolution imaging spectroradiometer (MODIS) land surface temperature (LST) data collected in this study.

| Study Area | Acquisition Time (Landsat 8 Data) | Acquisition Time (MODIS LST Data) |
|---|---|---|
| Jinan | 11 July 2014 02:48:23<br>25 April 2015 02:47:51<br>10 December 2017 02:48:40 | 11 July 2014 03:00:00<br>25 April 2015 02:55:00<br>10 December 2017 03:00:00 |
| Wuhan | 23 January 2014 02:57:26<br>24 July 2016 02:56:17<br>30 October 2017 02:56:36 | 23 January 2014 04:00:00<br>24 July 2016 03:05:00<br>30 October 03:05:00 |

### 2.2.1. MODIS Data

The MODIS LST product selected in this study is the MOD11_L2, collection 6 data with a spatial resolution of 1000 m; the MODIS LST data were downloaded from the NASA website (http://reverb.echo.nasa.gov/ accessed on 31 October 2020). The MOD11_L2, collection 6 product contained segmented data with a spatial resolution of 1000 m and was generated by the split-window algorithm. Extended validation works indicated that the accuracy of the MOD11_L2 product is approximately 1 K [43–45] and is widely used in LST downscaling algorithm research.

### 2.2.2. Landsat 8 Data

The Landsat 8 data of this study were downloaded from the geospatial data cloud (http://www.gscloud.cn/ accessed on 31 October 2020). The downloaded data were systematically processed with radiometric and geometric correction. In this study, the Landsat 8 data were processed with calibration and atmospheric correction modules from the ENVI 5.3 software to derive surface reflectance. In addition, accurate geometric correction between MODIS and Landsat 8 data was necessary. In this study, we used the image-to-image module of ENVI 5.3 software for geometric correction. The Landsat 8 data were selected as the reference images, and feature points such as road and river intersections were chosen as control points to correct MODIS images. Thus, the conjugated pixels from different sensor were matched and stacked to ensure the geometric consistency between images.

In this study, the MODIS LST was vital to model the relationship in the coarse spatial resolution. The Landsat 8 data serve two purposes in this study. One is to provide auxiliary parameters for establishing model, the auxiliary parameters including NDVI, SAVI, NDBI, UI and NDWI. These auxiliary parameters were retrieved from Landsat 8 data and resampled to 100 m and 1000 m for establishing the fine resolution and coarse resolution downscaling model, respectively. The resample method of spatial aggregations is adopted in this study to upscale the auxiliary parameters to coarse resolution of 100 m and 1000 m, so the values of coarse pixels are simply spatial means of the fine pixels. Another purpose is to provide validation reference data. The Landsat 8 LST was retrieved by the Mono-window algorithm and served as the validation data to verify and analyze the LST downscaling results.

The remote sensing inversion methods for LST are maturing and, the accuracy is constantly improving. Scholars have proposed different LST inversion methods for different remote sensing data. The common methods include a radiative transfer equation algorithm, Mono-window algorithm, and split-window algorithm [46–49]. MODIS LST usually uses the day/night algorithm and the split-window algorithm, whereas the Landsat series satel-

lites use the Mono-window algorithm with the support of in situ measured surface and atmosphere parameters. Compared with the radiative transfer equation, the Mono-window algorithm includes the influence of the surface and atmosphere in the calculation formula, making the application more convenient [50,51].

The United States Geological Survey (USGS) demonstrated that the calibration of the TIRS band 11 of Landsat 8 is temporarily unstable; therefore, the Mono-window algorithm, in combination with band 10 of Landsat 8, was used to retrieve LST data. The formula of the Mono-window algorithm is written as follows [52]:

$$T_S = \frac{a_{10}(1 - C_{10} - D_{10}) + [b_{10}(1 - C_{10} - D_{10}) + C_{10} + D_{10}]T_{10} - D_{10}T_a}{C_{10}} \quad (1)$$

where $T_S$, $T_{10}$ and $T_a$ represent land surface temperature, the brightness temperature of TIRS band 10, and effective average temperature of the atmosphere, respectively. Both $a_{10}$ and $b_{10}$ are constant. For band 10 of Landsat 8, the $a_{10}$ and $b_{10}$ can be seen in Table 2 [53].

**Table 2.** The value of $a_{10}$ and $b_{10}$.

| Range of LST (°C) | $a_{10}$ | $b_{10}$ | $R^2$ |
|---|---|---|---|
| 20–70 | 70.1775 | 0.4581 | 0.9997 |
| 0–50 | 62.7182 | 0.4339 | 0.9996 |
| −20–30 | 55.4276 | 0.4086 | 0.9996 |

The coefficient $C_{10}$ and $D_{10}$ can be calculated by the following equations:

$$C_{10} = \varepsilon_{10}\tau_{10} \quad (2)$$

$$D_{10} = (1 - \tau_{10})[1 + (1 - \varepsilon_{10})\tau_{10}] \quad (3)$$

where $\varepsilon_{10}$ and $\tau_{10}$ represent the land surface emissivity and atmospheric transmittance, respectively. It can be indicated that the effective atmospheric average temperature, atmospheric transmittance and surface emissivity are three significant parameters required when using the Mono-window algorithm to retrieve the land surface temperature. In general, the effective mean atmospheric temperature is obtained using the near-surface temperature by using a linear equation. Many factors influenced the atmospheric transmittance, including water vapor, aerosol, wavelength and ozone, although the atmosphere is the most important factor [54]. Two steps were used to estimate the surface emissivity. The first step used the land cover map to distinguish different land cover types in 30 m spatial resolution. The second step was the NDVI threshold method, used to estimate the surface emissivity [55].

## 3. Method

### 3.1. Introduction of Non-Linear Geographically Weighted Regressive Model

The relationships between LST and auxiliary parameters are complex and require in-depth exploration and research [56]. The development of the least square method starts from building linear relationship between LST and a single parameter to a linear relationship with multiple auxiliary parameters and further considering non-linear relationship. The traditional least square method ignores the non-stationarity in the relationship between the LST and auxiliary parameters. As an extension of the least square model, the GWR model is proposed as the typical model to consider the non-stationarity in downscaling LST; however, the non-linear relationship is ignored in the GWR model. In this study, we proposed a non-linear geographically weighted regressive (NL-GWR) model to address this issue. Compared with the GWR model, the NL-GWR added a non-linear auxiliary parameter in the process of building the model. The formula of NL-GWR is as follows:

$$y_i = \beta_0(u_i, v_i) + \beta_1(u_i, v_i)x_{i1}^2 + \beta_2(u_i, v_i)x_{i1} + \beta_3(u_i, v_i)x_{i2} + \ldots + \beta_n(u_i, v_i)x_{in} + \varepsilon_i \quad (4)$$

where, $y_i$ means dependent variable; $x_{i1}, x_{i2}, x_{in}$ represent different independent variables; $x_{i1}^2$ represents the quadratic of parameters, enabling the non-linear relationship between LST and auxiliary parameters; $n$ is the count of independent variables; $(u_i, v_i)$ represents the coordinate of the $i$th pixel. $\beta_0(u_i, v_i)$ refers to the intercept of the regression; the $\beta_1(u_i, v_i)$, $\beta_2(u_i, v_i)$ and $\beta_n(u_i, v_i)$ are the 1st, 2nd, and nth regression coefficients of the pixel, respectively. $\varepsilon_i$ is the random error. Fotheringham et al. used Tobler's first law of geography to determine the weight and proposed the weighted least square method to estimate the regression parameters [57]. The formula of regression parameters is as follows:

$$\hat{\beta}(u_i, v_i) = (X^T W(u_i, v_i) X)^{-1} X^T W(u_i, v_i) Y \tag{5}$$

where $\hat{\beta}$ is the estimated value of $\beta$; $X, Y$ are the vectors of independent and dependent variables, respectively; $W(u_i, v_i)$ represents the kernel function, which is used to ensure the weight of observation pixel. The weight of the observation pixel close to pixel $i$ will be larger, as the contribution of the observation pixel far away from pixel $i$ will be relatively small, the weight will also be relatively smaller. The kernel function selected in this study is obtained by the following formula:

$$W_{ij} = \exp(-\frac{d_{ij}^2}{b^2}) \tag{6}$$

where $d_{ij}$ is the Euclidean distance between the pixel $i$ and $j$; $b$ is the adaptive bandwidth, which can calculate by the cross-validation (CV) method of local regression analysis [58]. The relationship between CV and bandwidth can be shown as follows:

$$CV = \frac{1}{n} \sum_{i=1}^{n} [y_i - \hat{y}_{\neq i}(b)]^2 \tag{7}$$

where $\hat{y}_{\neq i}(b)$ means that the regression parameter estimation does not include the regression pixel itself and builds the relationship between the around pixels. When the relationship between the bandwidth and CV is built, the minimum CV corresponds to the optimal bandwidth.

### 3.2. LST Downscaling Algorithm Based on NL-GWR Model

In this study, we proposed the LST downscaling algorithm based on the NL-GWR model; a flow chart of the NL-GWR model downscaling LST algorithm is shown in Figure 2.

As shown in Figure 2, the NL-GWR model downscaling LST algorithm can be divided into three parts: data processing, LST downscaling model establishment, and verification and analysis of the downscaling results.

(1) Data processing. Firstly, the Landsat 8 reflectance data needed preprocessing, including radiometric calibration, atmospheric correction and geometric correction, etc., and then calculated the auxiliary parameters, including NDVI, SAVI, NDBI, UI, and NDWI using the Landsat 8 data, and resampled these indices to 1000 and 100 m, respectively. The data with a spatial resolution of 1000 and 100 m are the input parameters for fitting relationship in the coarse resolution and the fine resolution, respectively. For the MODIS LST data (MOD11_L2, collection 6 data), we used the MODIS reprojection tool (MRT) to registered to a UTM WGS 1984 reference system. In addition, then, the MODIS LST data were used to establish model in 1000 m resolution.

(2) LST downscaling model establishment. We used the coarse resolution auxiliary parameters and the MODIS LST to establish the NL-GWR model at a resolution of 1000 m, which is as follows:

$$\begin{aligned} LST_i^{CR} = \beta_0(u_i, v_i) &+ \beta_1^{CR}(u_i, v_i) index_{i1}^{2CR} + \beta_2^{CR}(u_i, v_i) index_{i1}^{CR} \\ &+ \beta_3^{CR}(u_i, v_i) index_{i2}^{CR} + \cdots + \beta_n^{CR}(u_i, v_i) index_{i(n-1)}^{CR} + \varepsilon_i^{CR} \end{aligned} \tag{8}$$

where the superscript *CR* denotes the data with a coarse resolution. $LST_i^{CR}$ is the MODIS LST at the pixel $i$; $index_{i1}^{2CR}, index_{i1}^{CR}, index_{i2}^{CR}$ and $index_{i(n-1)}^{CR}$ represent the quadratic and linear components of the auxiliary parameters at the pixel $i$, respectively. $\beta_1^{CR}, \beta_2^{CR}, \beta_3^{CR}, \beta_n^{CR}$ are the coefficient of quadratic component and coefficient of multiple one power parameters, respectively; $\varepsilon_i^{CR}$ is the fitting error at a coarse resolution.

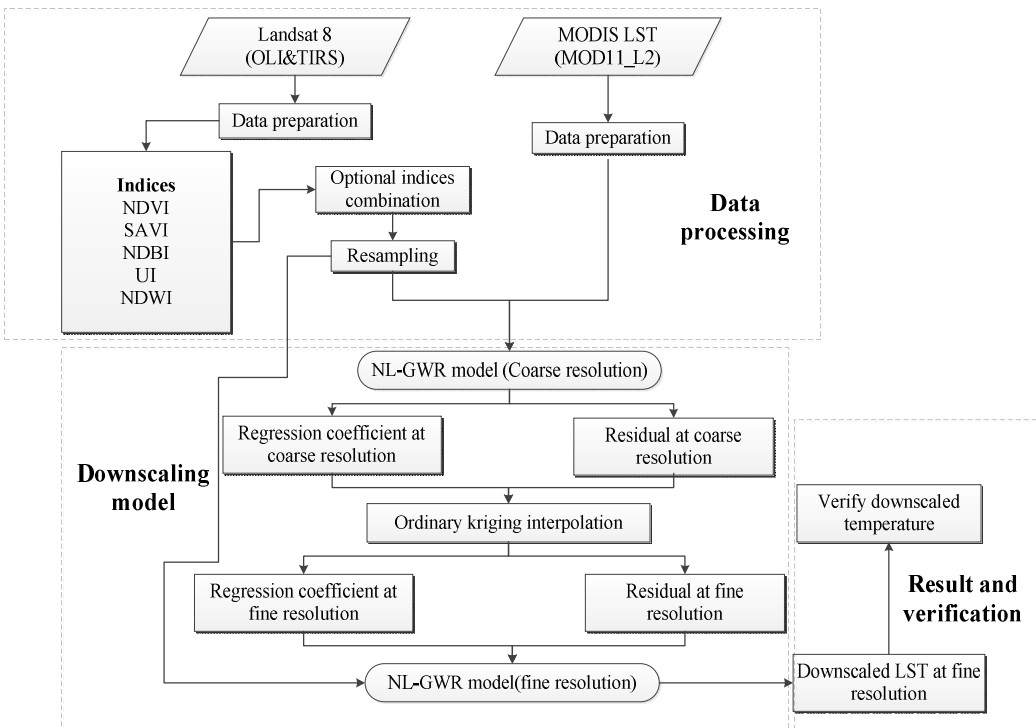

**Figure 2.** Flow chart of the non-linear geographically weighted regressive (NL-GWR) model downscaling LST algorithm.

Secondly, the coefficients and error are interpolated to 100 m, which was used to build a regressive relationship at the fine resolution in the Kriging interpolation module in ArcMap 10.2 software.

Finally, we assumed that the fitting relationship between LST and auxiliary parameters is irrelevant to the spatial resolution in the LST downscaling algorithm, which means the fitting relationship established for the coarse spatial resolution can be directly used for fine resolution modeling [37,59,60]; therefore, the relationship between LST and auxiliary parameters for the fine resolution can be expressed as follows:

$$
\begin{aligned}
LST_j^{FR} = \beta_0^{FR}(u_j, v_j) + \beta_1^{FR}(u_j, v_j)index_{j1}^{2FR} + \beta_2^{FR}(u_j, v_j)index_{j1}^{FR} \\
+ \beta_3^{FR}(u_j, v_j)index_{j2}^{FR} + \cdots + \beta_n^{FR}(u_j, v_j)index_{j(n-1)}^{FR} + \varepsilon_j^{FR}
\end{aligned} \tag{9}
$$

where the superscript *FR* represents the data at a fine resolution in Formula (9); the $LST_j^{FR}$ is the downscaled LST based on the NL-GWR model.

1.  Verification and analysis of the downscaling results. The LST from Landsat 8 retrieved by the Mono-window algorithm was used as reference data to verify the downscaled LST, and root mean square error (RMSE) and mean absolute error (MAE) were chosen as evaluating indicators. RMSE is the deviation between the observed value and its predicted value and illustrates the sample's dispersion degree. MAE represents the average value of the absolute error between the predicted value and the observed value. The smaller of RMSE and the MAE, the better the downscaling results.

## 4. Selection of Optimal Index

### 4.1. Candidates of the Remote Sensing Indices

NDVI can reflect the influence of the vegetation and eliminate the interference of soil and atmosphere [61]. The purpose of proposing SAVI is to correct the sensitivity of NDVI to the soil background. Compared with NDVI, SAVI adds the soil adjustment coefficient L into the formula, which is determined according to actual conditions. The range of SAVI is [0,1], and L = 1 means the vegetation coverage is high and the influence of soil background is low; L = 0 means the vegetation coverage is small. Normally, L = 0.5 can eliminate the influence of soil background, so L = 0.5 was selected in this study [62]. Zha et al. used the NDBI to describe the intensity of urbanization and distinguish building information. The range of NDBI is [−1,1], and areas of NDBI greater than 0 are considered urban land cover [63]. Urban index (UI) was proposed by Kawamura et al. to extract urban areas [64]. NDWI was used to extract water information in the study areas. NDWI is a measurement of liquid water molecules interacting with solar radiation [65] and was adopted in our study as an indicator of water information in the study area. The calculation formulas of the remote sensing indices used in this study are shown in Table 3.

**Table 3.** Remote sensing indices used and their calculation formulas.

| Indices | Abbreviation | Formulation | |
|---|---|---|---|
| Normalized Difference Vegetation Index | NDVI | $NDVI = \frac{R_{NIR} - R_{RED}}{R_{NIR} + R_{RED}}$ | (10) |
| Soil-Adjusted Vegetation Index | SAVI | $SAVI = \frac{(R_{NIR} - R_{RED})(1+L)}{R_{NIR} + R_{RED} + L}(L = 0.5)$ | (11) |
| Normalized Difference Built-up Index | NDBI | $NDBI = \frac{R_{SWIR1} - R_{NIR}}{R_{SWIR1} + R_{NIR}}$ | (12) |
| Urban index | UI | $UI = \frac{R_{SWIR2} - R_{NIR}}{R_{SWIR2} + R_{NIR}}$ | (13) |
| Normal Difference Water Index | NDWI | $NDWI = \frac{R_{GREEN} - R_{NIR}}{R_{GREEN} + R_{NIR}}$ | (14) |

In Table 3, $R_{NIR}$, $R_{RED}$, $R_{SWIR1}$, $R_{SWIR2}$ and $R_{GREEN}$ are the reflectance values of the near-infrared band, red band, the first shortwave infrared band, the second shortwave infrared band, and green band, respectively, and correspond to band 5, band 4, band 6, band 7, and band 3 of Landsat 8, respectively.

### 4.2. Optimal Index Combination of Research Areas

4.2.1. Downscaling with Single Remote Sensing Index

In this analysis, NDVI, SAVI, NDBI, UI, and NDWI were selected as auxiliary parameters to downscale the LST. The purpose of our first analysis was to choose the most sensitive parameter among the auxiliary parameters for building the downscaling relationship. For the selection of the optimal auxiliary parameters, we choose the stationary least square method that can explain the fitting accuracy of the auxiliary parameters and LST through the $R^2$ of fitting. We used the forward approach by successively adding significant terms into the model to select the optimal indices. The NDVI can reflect the influence of the vegetation and eliminate the interference of soil and atmosphere and is an important index for urban area LST downscaling. Therefore, we selected the NDVI as the first index, establish regressive relationship between Landsat LST and NDVI and calculate the $R^2$. Then, we added other index into regressive relationship and count $R^2$. At last, according to the $R^2$, we determined the optimal combination of auxiliary parameters. As all studied images have similar characteristics, one image for each city (Jinan: 11 July 2014 and Wuhan: 24 July 2016) was used to demonstrate the statistics of optimal index selection in Figures 3–7 and Tables 4–7.

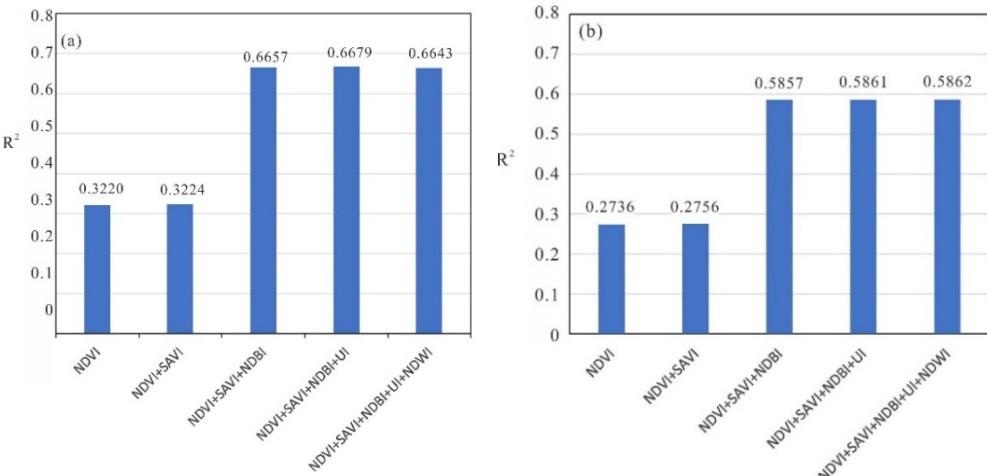

**Figure 3.** $R^2$ statistics of establishing regression relationship with different combination of indices. (**a**) and (**b**) are the $R^2$ statistics of Jinan (11 July 2014) and Wuhan (24 July 2016), respectively.

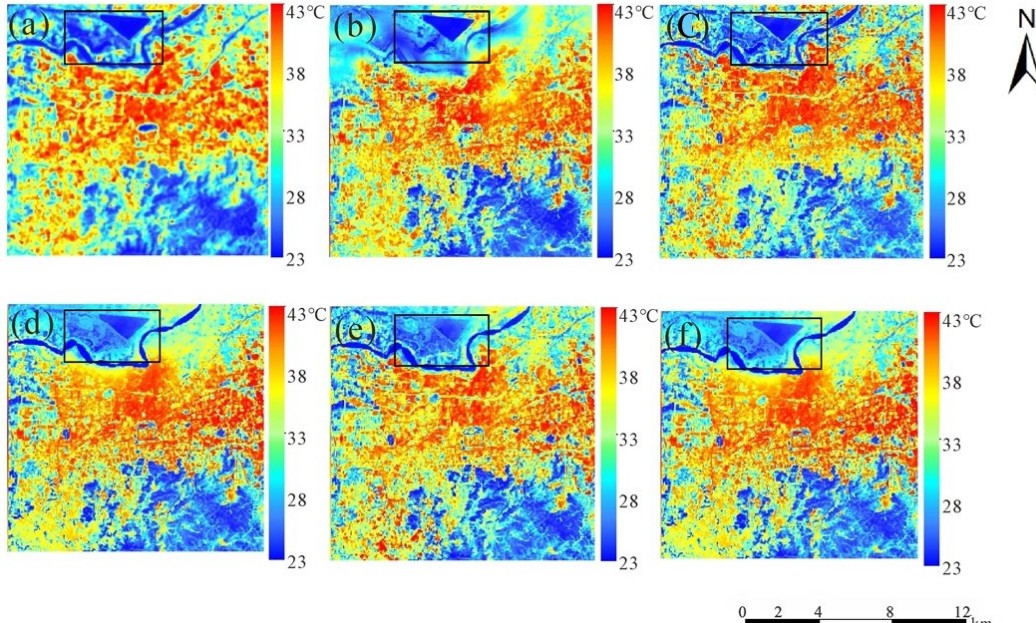

**Figure 4.** Downscaling results at a resolution of 100 m using different parameter combinations in Jinan (11 July 2014). (**a**) is the Landsat 8 LST retrieved by Mono-window algorithm; (**b**–**f**) are the downscaling results based on GWR and NL-GWR models and with the parameter combinations of NDVI + NDBI, $NDVI^2$ + NDBI, NDVI + $NDBI^2$, $NDVI^2$ + NDVI + NDBI, and $NDBI^2$ + NDBI + NDVI, respectively.

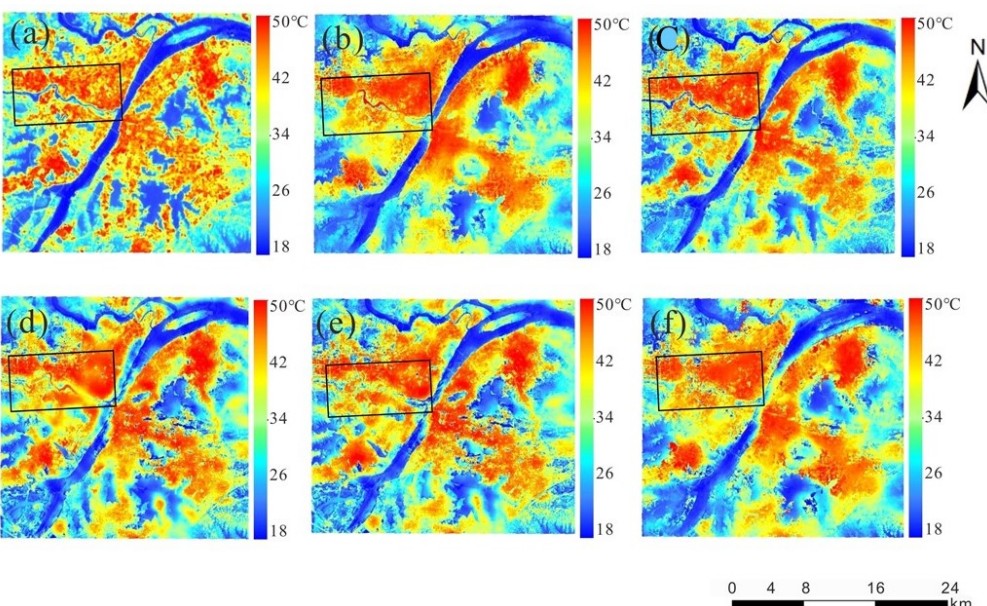

**Figure 5.** Downscaling results at a resolution of 100 m using different parameter combinations in Wuhan (24 July 2016). (**a**) is the Landsat 8 LST retrieved by Mono-window algorithm; (**b**–**f**) are the downscaling results based on GWR and NL-GWR models and with the parameter combinations of NDVI + NDBI, NDVI$^2$ + NDBI, NDVI + NDBI$^2$, NDVI$^2$ + NDVI + NDBI and NDBI$^2$ + NDBI + NDVI, respectively.

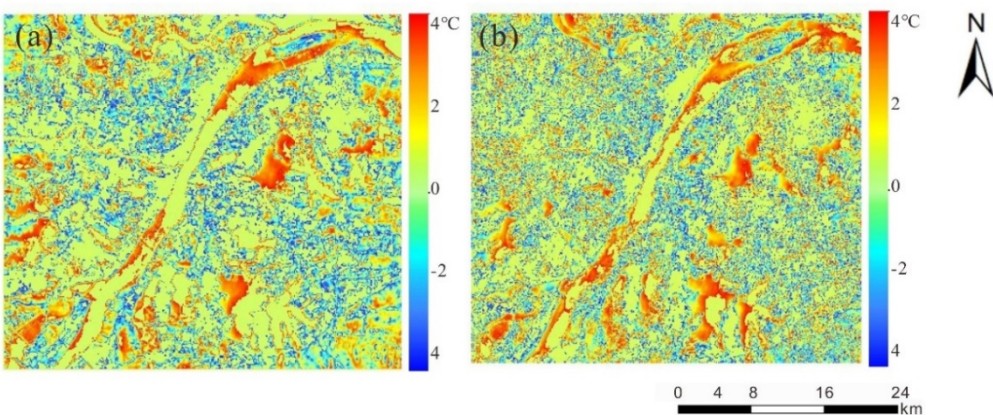

**Figure 6.** Error distribution between reference Landsat 8 LST and the downscaling results in Wuhan, 24 July 2016. (**a**,**b**) are the NL-GWR model results with NDVI$^2$ + NDBI and NDVI$^2$ + NDBI + NDWI parameter combinations, respectively.

As shown in Figure 3, the trend of the two areas is consistent. Compared with only using NDVI, when SAVI was added as index, the R$^2$ barely increased at all. Then, when NDBI was added as index, the R$^2$ has great improved, from 0.3224 to 0.6657 (Jinan, 11 July 2014) and from 0.2756 to 0.5857 (Wuhan 24 July 2016). Later on, When UI was added into the regressive relationship, the R$^2$ has little improvements. Finally, added NDWI into the regressive relationship, there were even a slight decrease in R$^2$. According to this analysis, SAVI, UI and the NDWI is not selected as the downscaling index. In total, we chose the NDBI, NDVI, and their quadratics as the auxiliary parameters in this study.

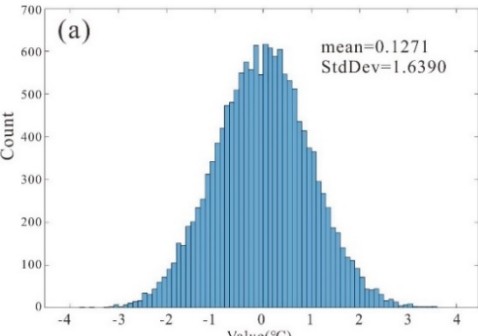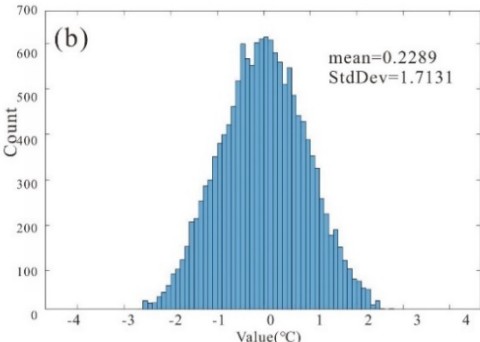

**Figure 7.** The histogram of error image corresponding to Figure 6. (**a**,**b**) are the error statistics of the parameter combination of NDVI$^2$ + NDBI and NDVI$^2$ + NDBI + NDWI.

### 4.2.2. Least Square Regression with Combined Remote Sensing Indices

Having analyzed each auxiliary parameter's sensitivity, we chose the optimal index combination of research areas. At first, we assumed that there are stationary relationships between the auxiliary parameters and LST and tested the performance of the combination of parameters in the sense of least square relationship. At a spatial resolution of 1000 m, we built a least square relationship between Landsat 8 LST and NDVI + NDBI, NDVI$^2$ + NDBI, NDBI$^2$ + NDVI, NDVI$^2$ + NDVI + NDBI, and NDBI$^2$+ NDBI + NDVI. The results of the two study areas are shown in Tables 3 and 4.

As shown in Tables 4 and 5, the two parameter combinations of NDVI$^2$ + NDVI + NDBI and NDBI$^2$ + NDBI + NDVI achieved better results in the global least square regression. The combination of NDVI$^2$ + NDBI also obtained better results in the Jinan area. In the global model, the fit results are better, obtain a better R$^2$, and lower RMSE when there are more parameters. Furthermore, the statistical results show that adding the quadratic of the auxiliary parameter can improve the least squares relationship's accuracy; therefore, it is necessary to consider the non-linear relationship between LST and auxiliary parameters.

**Table 4.** Least squares relationship results in Jinan (11 July 2014).

| Relationship | R$^2$ | RMSE (°C) |
|---|---|---|
| $LST = a + bNDVI + cNDBI$ | 0.66 | 1.38 |
| $LST = a + bNDVI^2 + cNDBI$ | 0.71 | 1.26 |
| $LST = a + bNDVI + cNDBI^2$ | 0.60 | 1.61 |
| $LST = a + bNDVI^2 + cNDVI + dNDBI$ | 0.75 | 1.15 |
| $LST = a + bNDBI^2 + cNDBI + dNDVI$ | 0.72 | 1.22 |

**Table 5.** Least squares relationship results in Wuhan (24 July 2016).

| Relationship | R$^2$ | RMSE (°C) |
|---|---|---|
| $LST = a + bNDVI + cNDBI$ | 0.42 | 0.96 |
| $LST = a + bNDVI^2 + cNDBI$ | 0.54 | 0.72 |
| $LST = a + bNDVI + cNDBI^2$ | 0.60 | 0.63 |
| $LST = a + bNDVI^2 + cNDVI + dNDBI$ | 0.70 | 0.59 |
| $LST = a + bNDBI^2 + cNDBI + dNDVI$ | 0.71 | 0.54 |

### 4.2.3. Geographical Weighted Regression with Combined Remote Sensing Indices

In fact, there is spatial non-stationarity between LST and the auxiliary parameters during the modeling process, which means the relationship will change when the geographic location changes [66]. In this section, we considered the non-stationary between auxiliary parameters to establish model. The GWR model takes the spatial location into

account in the modeling process that can reflect the independent variable's change rule on the dependent variable with the spatial location.

Tables 6 and 7 show the GWR and NL-GWR models' statistical results in the two study areas. As shown in Tables 5 and 6, the geographical model's statistical results are better than the least square model. Furthermore, by comparing Tables 3, 4, 5, and 6, we can see that adding the quadratic of the parameters into the model has a great effect on the improvement of results; therefore, considering the non-stationarity of the parameters with a geographical model and the non-linear relationship between LST and auxiliary parameters can improve the results, especially in the Wuhan area, where the coefficient of the determination of $R^2$ of the geographical model increased significantly.

**Table 6.** Geographical model results of Jinan (11 July 2014).

| Relationship | $R^2$ | RMSE (°C) |
|---|---|---|
| $LST = a(u_i, v_i) + b(u_i, v_i)NDVI + c(u_i, v_i)NDBI$ | 0.80 | 0.99 |
| $LST = a(u_i, v_i) + b(u_i, v_i)NDVI^2 + c(u_i, v_i)NDBI$ | 0.86 | 0.85 |
| $LST = a(u_i, v_i) + b(u_i, v_i)NDVI + c(u_i, v_i)NDBI^2$ | 0.81 | 0.98 |
| $LST = a(u_i, v_i) + b(u_i, v_i)NDVI^2 + c(u_i, v_i)NDVI + d(u_i, v_i)NDBI$ | 0.81 | 0.94 |
| $LST = a(u_i, v_i) + b(u_i, v_i)NDBI^2 + c(u_i, v_i)NDBI + d(u_i, v_i)NDVI$ | 0.83 | 0.89 |

**Table 7.** Geographical model results of Wuhan (24 July 2016).

| Relationship | $R^2$ | RMSE (°C) |
|---|---|---|
| $LST = a(u_i, v_i) + b(u_i, v_i)NDVI + c(u_i, v_i)NDBI$ | 0.60 | 0.76 |
| $LST = a(u_i, v_i) + b(u_i, v_i)NDVI^2 + c(u_i, v_i)NDBI$ | 0.82 | 0.39 |
| $LST = a(u_i, v_i) + b(u_i, v_i)NDVI + c(u_i, v_i)NDBI^2$ | 0.77 | 0.60 |
| $LST = a(u_i, v_i) + b(u_i, v_i)NDVI^2 + c(u_i, v_i)NDVI + d(u_i, v_i)NDBI$ | 0.79 | 0.53 |
| $LST = a(u_i, v_i) + b(u_i, v_i)NDBI^2 + c(u_i, v_i)NDBI + d(u_i, v_i)NDVI$ | 0.76 | 0.57 |

## 5. Results and Discussion

### 5.1. Downscaling Results

Figures 4 and 5 show the downscaling results at a spatial resolution of 100 m using different parameter combinations. Among them, Figures 4a and 5a are the Landsat 8 LST retrieved with the Mono-window algorithm applied to Jinan (11 July 2014) and Wuhan (24 July 2016), respectively; Figures 4 and 5b are the downscaling results based on GWR and NL-GWR models with the parameter combination of NDVI + NDBI, NDVI$^2$ + NDBI, NDVI + NDBI$^2$, NDVI$^2$ + NDVI + NDBI, and NDBI$^2$ + NDBI + NDVI, respectively. Figures 4 and 5b are the traditional GWR model, whereas Figures 4 and 5c–f added the quadratic of the parameters considering the non-linear relationship between LST and auxiliary parameters.

As shown in Figure 4, the downscaling results with different parameter combinations are consistent with the reference Landsat 8 LST visually speaking. Compared to the NL-GWR model, the GWR model results only consider the linear relationship between the LST and NDVI + NDBI parameter combination, which is substandard in the Yellow River passing through in the north of the Jinan area. The LST of the water is higher than the reference LST because there is a negative linear correlation between the LST and NDVI in the area with high vegetation coverage; however, the NDVI and LST is not a linear relationship in the area with abundant soil moisture. In the previous studies of geographical models, the non-linear relationship of the heterogeneous urban areas and areas with abundant soil water content is neglected, which caused large downscaling errors. The difference is quite large in the north of the study area, where a reservoir is marked with a black rectangle in Figure 4, and different combinations of parameters have different results around the reservoir. As shown in the Landsat 8 LST (Figure 4a), the LST of the reservoir is the lowest, but the LST of downscaling results in Figure 4d,e is slightly higher.

From the visible results, the NL-GWR model's result with the parameter combination $NDVI^2$ + NDBI is closest to the reference data and shows the best visible results.

As Figure 5 shows, Wuhan has a similar visual effect to Jinan. The downscaling image trend is consistent with the reference image, but the downscaled LST has a certain smoothing effect in some areas. Two possible reasons can explain this phenomenon. One reason could be that the processes of regression and interpolation are both based on the minimum mean square error (MMSE) method. There is a common problem for the MMSE method, i.e., that low values tend to be overestimated and high values tend to be underestimated. Another reason is that, in the process of spatial aggregation from 100 to 1000 m of NDBI and NDVI, values of NDVI and NDBI will be replaced by the average values of surrounding pixels, making the resampled image smoother than the original image; this may cause the smoothing effect of the downscaling results. The downscaling LST is the highest in the Yangtze River's tributary when the NDVI + NDBI is used as the auxiliary parameter combination (marked with a black rectangle in the figure). This situation is not consistent with the reference image, which causes a relatively significant error. By considering the non-linear relationship between LST and auxiliary parameters, the LST of the tributaries has decreased. In the visual images, the downscaling results of the NL-GWR downscaling model with the $NDVI^2$ + NDBI parameter combination are closest to the reference image; this result is consistent with the Jinan area.

In the Wuhan study area, the Yangtze River runs through it, and there are many lakes. Given that there is much water in and around Wuhan, it is intuitive to add the NDWI index into the downscaling model; therefore, the NDWI was added into the optimal parameter combination $NDVI^2$ + NDBI to downscale the LST and verify the accuracy of the downscaling results. The downscaling results of $NDVI^2$ + NDBI and $NDVI^2$ + NDBI + NDWI downscaling were evaluated by analyzing their errors. Figure 6a,b show the error distributions between reference Landsat 8 LST and the downscaling results of the NL-GWR model with $NDVI^2$ + NDBI and $NDVI^2$ + NDBI + NDWI as parameter combinations, respectively. The error range of Figure 6a,b is –4~4 °C. In Figure 6a, the large errors are mainly in rivers, lakes, and high soil water content, and the intersecting areas of different land cover when using the $NDVI^2$ + NDBI parameter combination. Though the errors of using the $NDVI^2$ + NDBI + NDWI parameter combination decreased in the lake, river, and high soil water content areas, in the vegetation, building and other areas were increased, as shown in Figure 6b.

Figure 7 shows the histogram of the error image, and Figure 7a,b correspond to Figure 6a,b. The number of pixels in the interval of $[-1,1]$ in Figure 7a is higher than that in Figure 7b. We selected the mean value and standard deviation (StdDev) as evaluation indicators. The mean value was calculated based on all pixels and is preferably as close to 0 as possible, which indicates unbiased prediction. By comparing Figure 7a,b, we found that the mean values are 0.1271 and $-0.2289$ °C and the StdDev values are 1.6390 and 1.7131, respectively. So, the mean value of Figure 7a is closer to 0, and the StdDev of Figure 7a is smaller than (b), which means that the dispersion is also smaller.

In summary, adding the NDWI into the downscaling model can improve the downscaling results of water areas; however, the errors of the other land covers have increased. The downscaling results with $NDVI^2$ + NDBI have a larger error in the water areas, but the overall accuracy is higher than that of adding the NDWI and obtaining better downscaling results. The conclusion is that NDWI is not needed in LST downscaling applications.

The visual results show that the downscaling results are improved when considering the non-linear relationship between LST and auxiliary parameters. It is necessary to consider the non-linear relationship in the geographical downscaling model. In the two study areas, the parameter combination of $NDVI^2$ + NDBI obtained the best visual results, and adding the NDWI into the parameter combination does not improve the error statistics of the NL-GWR model.

## 5.2. Landsat 8 LST as the Reference Data

It is necessary to quantitatively verify the downscaling effect of different parameter combinations instead of relying on simple visual inspection. Ideally, the land surface temperature detected in the ground is the most accurate and direct source for verification data; however, LST cannot accurately be measured from the ground due to its extreme variability as well as scale effect, not to mention the impractical workload to collect representative samples for the whole study area. As a compromise, the inversed high-resolution LST from Landsat 8 data is usually selected as the reference data in the LST downscaling studies.

Significant amounts of research have shown that if there is no ground monitoring data, the LST accuracy retrieved by the MODIS can be used as a reference [67–69]. Peng and wang et al. proved that the relationship between the MODIS LST and the LST retrieved by the Landsat 8 is preferable and that the RMSE is less than 2 °C [41,42]. Duan et al. used the ASTER LST as reference data to evaluate the downscaling LST based on the GWR model and demonstrated an error of 2.1 K for the ASTER LST and MODIS LST, which is bigger than the RMSE found between MODIS LST and Landsat 8 retrieved LST at a 1000 m spatial resolution [39]. For this reason, the Landsat 8 LST retrieved by the Mono-window algorithm can be used as reference data to verify the accuracy of downscaling results.

In this study, the Landsat 8 LST was used as reference data to verify the downscaling LST, with RMSE and MAE used as indicators for verification. The statistics of RMSE and MAE of Jinan and Wuhan are shown in Tables 8 and 9, respectively. The results are based on the GWR and NL-GWR models with different parameter combinations.

As shown in Tables 8 and 9, the following conclusion can be drawn from the six datasets: the quantitative analysis results are consistent with the visual results; the NL-GWR model with the parameter combination of NDVI2 + NDBI obtains the best downscaling LST. In all the datasets, RMSE and MAE are smaller when considering the non-linear relationship between LST and auxiliary parameters. In the perspective of seasonality, the summer data achieved better downscaling results than other seasons and gained the smallest RMSE (Jinan (11 July 2014): 1.3208 °C, Wuhan (24 July 2016): 1.4957 °C) and MAE (Jinan: 0.9208 °C, Wuhan: 1.0686 °C). The statistical results show that consideration of the non-linear relationship improves the accuracy of downscaling results.

**Table 8.** Accuracy statistics of downscaling results in Jinan.

| Parameter Combination | 11 July 2014 | | 25 April 2015 | | 10 December 2017 | |
|---|---|---|---|---|---|---|
| | RMSE | MAE | RMSE | MAE | RMSE | MAE |
| NDVI + NDBI | 2.4997 | 1.4054 | 2.5516 | 1.4687 | 2.3328 | 1.3574 |
| NDVI$^2$ + NDBI | 1.3208 | 0.9208 | 1.4858 | 0.9721 | 1.5231 | 1.0063 |
| NDVI + NDBI$^2$ | 1.8926 | 1.2451 | 1.7976 | 1.1526 | 2.1810 | 1.2762 |
| NDVI$^2$ + NDVI + NDBI | 2.0224 | 1.3708 | 2.1364 | 1.3840 | 2.3180 | 1.4308 |
| NDBI$^2$ + NDBI + NDVI | 1.6921 | 1.1764 | 2.1221 | 1.2765 | 2.3100 | 1.2364 |

**Table 9.** Accuracy statistics of downscaling results in Wuhan.

| Parameter Combination | 23 January 2014 | | 24 July 2016 | | 30 October 2017 | |
|---|---|---|---|---|---|---|
| | RMSE | MAE | RMSE | MAE | RMSE | MAE |
| NDVI + NDBI | 2.9117 | 2.1017 | 2.8165 | 1.9757 | 2.0343 | 1.4837 |
| NDVI$^2$ + NDBI | 1.7235 | 1.2384 | 1.4957 | 1.0686 | 1.4168 | 1.1172 |
| NDVI + NDBI$^2$ | 2.5611 | 2.1609 | 3.0211 | 1.1524 | 1.6821 | 1.2407 |
| NDVI$^2$ + NDVI + NDBI | 2.5039 | 2.1436 | 2.0224 | 1.2844 | 1.8566 | 1.2280 |
| NDBI$^2$ + NDBI + NDVI | 2.5445 | 2.2140 | 2.5254 | 1.6298 | 1.7975 | 1.2451 |

Figures 8 and 9 show the density scatter plots between the reference Landsat 8 LST and the downscaling LST-based NL-GWR model with the parameter combination of NDVI$^2$ + NDBI in the six pairs of study images with a 100 m spatial resolution.

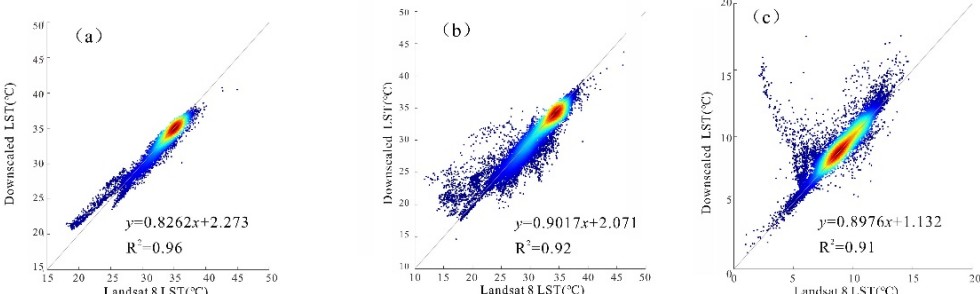

**Figure 8.** Density scatter plots between the reference Landsat 8 LST and the downscaling LST with a spatial resolution of 100 m in Jinan: (**a**) 11 July 2014; (**b**) 25 April 2015; (**c**) 10 December 2017.

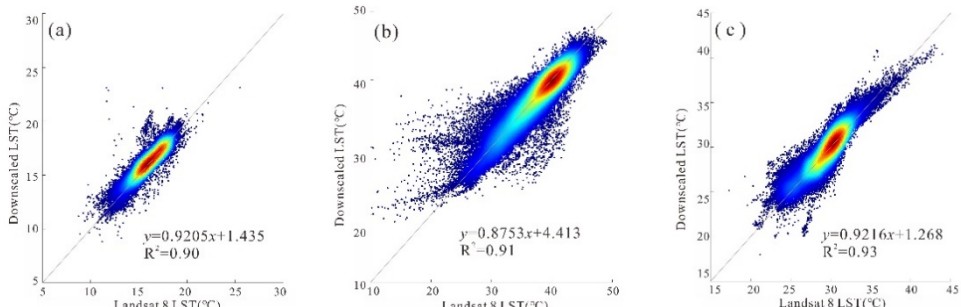

**Figure 9.** Density scatter plots between the reference Landsat 8 LST and the downscaling LST with a spatial resolution of 100 m in Wuhan: (**a**) 23 January 2014; (**b**) 24 July 2015; (**c**) 30 October 2017.

Figures 8 and 9 show that all fitting coefficients of determination are above 0.90 and obtain satisfactory fitting results, indicating that the NL-GWR model can achieve better downscaling results than the GWR model and improve the accuracy of the downscaling algorithm. This conclusion provides new approaches to the study of downscaling algorithms.

## 6. Conclusions

Due to the spatial heterogeneity of the urban area's underlying surface, it is necessary to gain LST data with high spatial and temporal resolutions; however, high spatial and temporal resolution can not be simultaneously achieved by current satellite sensors. To solve this problem, we proposed an NL-GWR model, which simultaneously considers the spatial non-stationarity and non-linearity in order to downscale the spatial resolution of MODIS LST from 1000 to 100 m and verify the downscaling results comparing with the Landsat 8 retrieved LST.

In this study, Jinan and Wuhan were selected as study areas. We selected the auxiliary parameters among the NDVI, SAVI, NDBI, UI, and NDWI, and chose the NDVI and NDIBI as auxiliary parameters; then, we selected the different combinations of NDVI + NDBI, $NDVI^2$ + NDBI, NDVI + $NDBI^2$, $NDVI^2$ + NDVI + NDBI, and $NDBI^2$ + NDBI + NDVI to downscale LST; lastly, by comparing and verifying the downscaling results from different parameter combinations, we concluded that the NL-GWR model with the $NDVI^2$ + NDBI parameter combination obtained the best downscaling results. Due to Wuhan's many water bodies, we tried to add NDWI into the downscaling parameter combination. The downscaling results were improved for areas containing water, but the error in other areas increased; therefore, the NDWI was not used as an auxiliary parameter in this study. The retrieved Landsat 8 LST by the Mono-window algorithm was used as reference data to verify the LST downscaling results. In six datasets, the coefficients of determination between downscaling LST and the Landsat 8 LST both reached above 0.90.

The downscaling algorithm based on the NL-GWR model has achieved better downscaling results than the former GWR model, which only considers linear relationships.

Although the NL-GWR model proposed in this study has achieved good downscaling results, many problems still need to be improved in subsequent studies. In this study, only the auxiliary parameters' quadratic parameters were added to the model the non-linear relationship between LST and auxiliary parameters, although more non-linear relationships, e.g., exponential or rational functions, could be considered. Another aspect that needs improvement is that more indices besides NDVI, SAVI, NDBI, UI, and NDWI should be considered to adapt to the urban land cover diversities. These problems need to be further explored in the future studies.

We consider our study a successful attempt to build a non-linear relationship in geographically weighted regressive algorithms where the downscaling result's accuracy is significantly improved due to the introduction of non-linear terms. Two cities with different climate, as well as different seasonal data, were investigated, and the proposed method gets better downscaling results in all datasets. Therefore, we believe that the NL-GWR model can be applied to other urban area around the world. However, in extremely heterogeneous areas such as mountains, using nonlinear term in LST downscaling may introduce unstable result and may need extra constraints. It is also recommended that before applying this algorithm to scenarios other than urban areas, selection of more auxiliary parameters and optimization of their combination should be performed. For example, slope angle or incoming solar radiation may be more relevant in mountainous areas than NDBI. The overall procedure was repeatable, and the data are available for many different practices, thus providing a new urban heat island study tool.

**Author Contributions:** Conceptualization, Q.L. and X.L. (Xiaobo Luo); methodology, S.W.; soft-ware, Y.L. and K.Y.; validation, X.L. (Xia Li), Y.L. and X.L. (Xiuhong Li); writing—original draft preparation, Q.L. and S.W.; writing, review and editing, S.W., Q.L., Y.L., X.L. (Xia Li) and K.Y. All authors have read and agreed to the published version of the manuscript.

**Funding:** This research was funded by the National Natural Science Foundation of China (No. 20180913), the National Key Research and Development Program of China (No. 2020YFA0608703), and Research on comprehensive management and system innovation of water environment quality in Beijing Tianjin Hebei region (No. 2018ZX07111).

**Acknowledgments:** The authors would like to thank the contributions of the anonymous reviewers and the Institute of Remote Sensing and Digital Earth Chinese Academy of Sciences. The authors would also like to thank NASA for providing the satellite data and insightful comments that helped significantly improve this manuscript.

**Conflicts of Interest:** The authors declare no conflict of interest. The funders had no role in the design of the study; in the collection, analyses, or interpretation of data; in the writing of the manuscript, or in the decision to publish the results.

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
