# Peer review of "Downscaling Land Surface Temperature Based on Non-Linear Geographically Weighted Regressive Model over Urban Areas"

_remotesensing, doi:10.3390/rs13081580_

Round 1

Reviewer 1 Report

Review of " Downscaling land surface temperature based on non-linear geographically weighted regressive model over urban areas" by Wang et al. for Remote Sensing.

In study MODIS LST is downscaled based on a non-linear geographically weighted regressive (NL-GWR) model. NDVI and NDIBI are selected as auxiliary parameters (among NDVI, SAVI, NDBI, UI, and NDWI), and then different combinations of these two are applied to downscale LST. To test the results, they used Landsat 8 LST as reference. They show that the NL-GWR model performed better than linear approaches in both studies areas. The topic (downscaling LST approaches and their validation) and overall content of the article is of great interest for the journal readers.

The article is a resubmission in which the serious flaws of the former version in both content and form have been resolved. The article is now written and structured in an appropriate way, the data and analyses are much better presented and unnecessary information and equations have been removed.

The most remarkable improvement content-wise is the fact that they now evaluate the proposed NL-GWR model with images from different years and seasons. However, I still miss in the conclusion a paragraph explaining whether this method could be applied to other urban areas around the world and in that case, what are the restrictions and limitations?

Minor revision is advised prior to publication addressing the specific comments indicated in the attached file.

Reviewer 2 Report

The paper describes a process for selecting an optimal combination of parameters that take part in a non-linear geographically weighted regressive (NL-GWR) model to downscale the spatial resolution in the (MODIS) LST products to the finest resolution in Landsat 8 data. An assumption of scale invariability of the relationships between LST and different common remote-sensing indices is made. Although it is written straightforward, the text is sometimes very hard to follow, and the reader misses the natural flow it should contain.

Therefore this reviewer has met some concerns that have to be addressed:

Why assigning the term “global” model, fitting, regression, etc, when it is used for a satellite scene only? If this is the case, another term should be selected.

Alternatively, the term “localized” model or relationship is not properly defined. From the text, it comes off that the term is closely related to “non-stationarity”, another term that is vaguely defined in line 383, a bit late for the reader. Although the term “geographical” has also been introduced, in the case the concept of “localized” is used as synonymous with “geographical”, why is this last not used instead? There is a good number of papers in your reference list that are entitled to the term “geographical” but none of them with the term “localized”. Please consider making a change in your terminology.

Table 1: Besides the scene date as you provide in the table, could you specify acquisition hour and minute?

Line 216: Could you provide more information on the geometric correction procedure to relate Landsat 8 pixels to the MODIS ones?

Line 218-219: Please provide more explanations in a synthesized way to clarify the Landsat 8 data utility.

Lines 286-291: Please, as the processing of Landsat 8 reflectance data and MODIS LST data during the data processing section runs into parallel flows but not mixed, provide a separate explanation for each type of processing.

Equation 8: At the coarse resolution, it denotes different coefficients in the equation for each of the MODIS LST pixels. How can you obtain so many equations when you only have a set of MODIS LST points and a corresponding set of indices? I cannot see so many points as you claim.

Lines 300-302: The sentence is hard to follow if the above comment applies.

Section 4.2: To obtain the model equations, alternatively I would propose the use of a stepwise regression in which an automatic procedure is in charge to choose the most predictive variables. There are three types of operative steps: forward approach by successively adding significant terms into the model, backward elimination by subtracting irrelevant terms from a starting model that considers all the statistical terms initially, and finally bidirectional elimination as a combination of both. Each introduced significant term may explain a percentage of the LST difference variability, which can be represented as 100*R2.

Please, rewrite the whole text to make it clearer to the reader.

Round 2

Reviewer 2 Report

This reviewer appreciates the efforts made by the authors in order to clarify and improve the manuscript. I see this study closer to publication. However, some concerns are still to be clarified. I suggest the authors put themselves into the readers' role and make the necessary changes accordingly.

Line 162: “geographical GWR” repeats twice the term geographical. Much better if you simply leave “GWR”.

Table 1: Acquisition times for the MODIS LST data coincide to the second. Obviously, this reviewer is not going to confirm those provided times, but for a normal reader, this sounds awkward. Acquisition times should correspond to the center of each scene, which presumably would match the centers of Jinan and Wuhan cities.

Line 221: You better clarify how Landsat 8 data were resampled to 100m and 1000m resolutions, most appropriately for the coarse resolution, in order to make it clearer to the reader if coarse pixels are simply spatial means of the fine pixels.

Equation 8: A better explanation for the reason why “each pixel in the MODIS LST data corresponds to a regression equation” should be provided right in the manuscript. If this comes from the “explanations of the GWR model”, then a paraphrase introducing this aspect should be provided, maybe in the methodology section. Therefore, equation 8 is not fully understood if the quadratic and linear components of the auxiliary parameters index1, index2, …, indexn-1; are not written as indexi1, indexi2, …, indexin-1; referring ultimately to pixel i.

Equation 9: The same as above applies to this equation. As this equation is for the finest resolution, pixels do not correspond to the ones at coarse resolution. Therefore, I would not advise using sub-index i for this resolution again, maybe j would be perfect. And, similarly, as above, index1, index2, …, indexn-1 should be denominated now as indexj1, indexj2, …, indexjn-1.

Equations 8 and 9 are basic in this paper. If they are not properly defined and expressed, then the reader is not going to understand the rest of the manuscript, making him/her reread previous parts over and over again.

Section 4.2: In this section, there are 3 subsections. Each of them assumes different conditions in terms of data stationarity. This should be clearly addressed in each of the subsections: where stationarity or non-stationarity are assumed. The reader will appreciate this.

Figures 4 and 5: They are the same, so this must be a mistake.

Author Response

This manuscript is a resubmission of an earlier submission. The following is a list of the peer review reports and author responses from that submission.

Round 1

Reviewer 1 Report

Im happy with the quality of writing and the technical aspect of the article as its well written and described. 

Reviewer 2 Report

Review of "Downscaling land surface temperature based on non-linear geographically weighted regressive model over urban areas" by Wang et al. for Remote Sensing.

This study proposes downscaling MODIS LST based on a non-linear geographically weighted regressive (NL-GWR) model. NDVI and NDIBI are selected as auxiliary parameters (among NDVI, SAVI, NDBI, UI, and NDWI), and then different combinations (NDVI+NDBI, NDVI2+NDBI, NDVI+NDBI2, NDVI2+NDVI+NDBI and NDBI2+NDBI+NDVI) are applied to downscale LST. To test the results, they used Landsat 8 LST as reference. They conclude that NL-GWR model with the parameter combination of NDVI2+NDBI provide the best downscaling results.

Although the topic (downscaling LST approaches and their validation) and overall content of the article is of great interest for the journal readers, the article has serious flaws in both content and form. In my opinion it does not meet the quality standards and acceptance criteria of the journal. I encourage authors to resubmit their work once they have dealt with the following issues.

The article is not written or structured in an appropriate way (I have flagged many mistakes in the attached file) and the data and analyses are not well presented. The authors recently published a closely related paper: DOI: 10.1109/JSTARS.2020.2968809, which is much better written and presented (please see my comments in the abstract and Figure 1 in attached file).

As regards contents, the introduction is not up to date and does not include closely related studies (on downscaling MODIS LST) from 2020 such as DOI: 10.3390/rs12132134, DOI: 10.1016/j.isprsjprs.2020.01.014, DOI: 10.1109/ACCESS.2020.3021034. The methodology is the weakest part of the article and major improvement is required to meet the journal standards. Only two images (July 11, 2014 and July 24, 2016 for Jinan and Wuhan, respectively) are used to evaluate the proposed NL-GWR model, and there is no justification provided for that choice. In that sense: (i) Why only two images if there are plenty available? (ii) Why are the features of these images good enough to evaluate the model? (iii) Can this method be applied to other urban areas (worldwide)? (iv) What are the restrictions and limitations? (v) Using only two images can lead to bias in your evaluation results (spatial and temporal -wise) (vi) I agree with your statements in lines 575-581, so you may address some of them in the present study and provide a more comprehensive analysis. Apart from these major concerns, the manuscript includes unnecessary equations like the calculation of RMSE and MAE (eq. 10 and 11), defines basic vegetation Indexes (like NDVI) well known for “remote sensing” readers and seems to have left part of the author guidance of the template in the text (lines 309-311). The results and conclusion also need a makeover. I have flagged some aspects in the attached file to provide guidance to the authors.

Reviewer 3 Report

The study aims at using the NL-GWR model to reduce the spatial resolution of the MODIS LST from 1000 m to 100 m and to verify the downscaling results by the LST retrieved by Landsat 8. The idea is somewhat original and it will allow having LSTs from low spatial resolution MODIS images covering a large area, almost identical to that obtained by Landsat. 

The authors have well stated the abstract as well as the Introduction. The methodology is well illustrated and the results are well presented and evaluated.

 However, some sections are not useful for the paper, particularly the two paragraphs of the 4th section. I recommend to the authors to summarize them by adding some references related to the different indices.